# Prevalence of Plasmid-Mediated Quinolone Resistance (PMQRs) Determinants and Whole Genome Sequence Screening of PMQR-Producing *E. coli* Isolated from Men Undergoing a Transrectal Prostate Biopsy

**DOI:** 10.3390/ijms23168907

**Published:** 2022-08-10

**Authors:** Katarzyna Piekarska, Katarzyna Zacharczuk, Tomasz Wołkowicz, Rafał Gierczyński

**Affiliations:** Department of Bacteriology and Biocontamination Control, National Institute of Public Health NIH—National Research Institute, 24 Chocimska Str., 00-791 Warsaw, Poland

**Keywords:** WGS 2, Post-biopsy infections, *Escherichia coli*, PMQR 5, virulence factors

## Abstract

Fluoroquinolones (FQs) are recommended as prophylaxis for men undergoing transrectal prostate biopsy (TRUS-Bx). Recent studies suggest a significant share of FQ-resistant rectal flora in post-TRUST-Bx infections. Methods: 435 *Enterobacterales* isolates from 621 patients attending 12 urological departments in Poland were screened by PCR for PMQR genes. PMQR-positive isolates were tested for quinolone susceptibility and investigated by whole genome sequencing (WGS) methods. Results: In total, 32 (7.35%) *E. coli* strains with ciprofloxacin MIC in the range 0.125–32 mg/L harbored at least one PMQR gene. *qnrS* and *qnrB* were the most frequent genes detected in 16 and 12 isolates, respectively. WGS was performed for 28 of 32 PMQR-producing strains. A variety of serotypes and sequence types (STs) of *E. coli* was noticed. All strains carried at least one virulence gene. AMR genes that encoded resistance against different classes of antibiotics were identified. Additionally, five of 13 ciprofloxacin-susceptible *E. coli* had alterations in codon 83 of the GyrA subunits. Conclusion: This study provides information on the common presence of PMQRs among *E. coli*, which may explain the cause for development of post-TRUS-Bx infections. High numbers of virulence and antimicrobial resistance genes detected show a potential for analysed strains to develop infections.

## 1. Introduction

Fluoroquinolones (FQs) are an important class of synthetic broad-spectrum antibacterial agents used in medicine for treatment of multiple infections. However, their increased use has led to increasing bacterial resistance, mainly among different species of *Enterobacterales*, creating a challenge for the effectiveness and safety of therapies used against infections caused by these bacteria.

Recently, the emerging threat of infectious complications after transrectal ultrasound-guided prostate biopsy (TRUS-Bx) has been reported [1,2,3]. The most common pathogen in the setting of post-biopsy infectious complications is *Escherichia coli*, with causative strains probably originating from the patient’s own endogenous flora [4]. Moreover, the prevention and management of TRUS-Bx infectious due to *E. coli* has become more complicated due to widespread emergence of FQ-resistant *E. coli* isolates. Such strains may be selected as a result of any FQ therapy or by FQ prophylaxis conventionally used prior to biopsy [5]. Studies suggest that pre-biopsy screening for such FQ-resistant pathogens, with subsequent “tailored” prophylaxis based on antimicrobial susceptibility results, may be an effective way to reduce infectious complications [1,6,7]. Therefore, there is a growing need to understand FQ-resistant strains by monitoring their prevalence and identifying predicting factors.

The main mechanism conferring high-level FQ resistance is the result of spontaneous point mutations within DNA gyrase and topoisomerase IV coding target enzymes for quinolones. A point mutation results in amino acid substitutions within the quinolone resistance determining region (QRDR) of *gyrA* and/or *parC* chromosomal genes encoding subunit A gyrase (GyrA) and subunit C (ParC) topoisomerase IV, respectively [8]. Although considerably less frequent, mutations in subunit B (GyrB) of DNA gyrase and subunit E (ParE) of topoisomerase IV have also been found to confer resistance to FQ. Meanwhile, in recent decades, plasmid-mediated quinolone resistance (PMQR) among *Enterobacterales* has become a global phenomenon [9]. PMQR mechanisms encoded by plasmid genes include Qnr peptides (capable of protecting DNA gyrase and topoisomerase IV from quinolones), aminoglycoside acetyl transferase *aac*(6′)-*ib*-*cr* (modifying quinolones with a piperazinyl substituent), and the quinolone efflux pumps—QepA and OqxAB [9]. The co-existence of mutations in QRDR and PMQR determinants in FQ-resistant *Enterobacterales* was described in previous works [9,10,11]. However, in the absence of chromosomally mediated quinolone resistance mechanisms (mutations), the acquisition of plasmid-mediated quinolone resistance mechanisms (PMQRs) leaves fluoroquinolones’ MIC values in the susceptible category, according to the EUCAST breakpoints criteria. It should also be noted that the presence of PMQRs has been found to promote QRDR mutations, increasing FQ resistance rates [9]. In addition, PMQRs transmitted on plasmids are capable of rapid spread between bacteria of different species, therefore giving rise to potential complications for the use of fluoroquinolones as therapy or prophylaxis. Consequently, the presence of PMQR-positive bacteria within rectums of patients undergoing biopsy are considered an important risk factor for complications and may pose a serious challenge to the treatment of infections.

Some studies have previously investigated the range of post-TRUS-Bx complications in the context of FQ resistance. Recently, our team published a study on the molecular mechanisms of FQ-resistance including the *gyrA*, *gyrB* and *parC*, *parE* mutations and PMQRs of *Enterobacterales* from rectal swabs of 48 men undergoing transrectal prostate biopsy [12]. Until recently, little has been known about the molecular properties of *E. coli* as causative agents of post-biopsy infections. Thus, we aimed to determine the prevalence of PMQR determinants in *Enterobacterales* strains isolated from rectal swabs of 621 men undergoing transrectal prostate biopsy at 12 urological hospitals in Poland. Then, we investigated PMQR-producing *E. coli* strains by WGS for antibiotic resistance genes, virulotype, and genotype, that potentially could be responsible for infections associated with post-TRUS-Bx complications in patients in Poland.

## 2. Results

### 2.1. Bacterial Isolates

A total of 435 *Enterobacteriaceae* strains were cultured from 621 rectal swabs of men undergoing prostate biopsy procedure, collected at 12 urological departments participating in the presented study. *E. coli* was the most prevalent gram-negative bacteria isolated. The main criterion for the selection of *E. coli* isolates for testing was the presence of PCR-product of the most commonly PMQRs i.e., *qnrA*, *qnrB*, *qnrS*, *aac(6*′*)-Ib-cr*, and *qepA*.

### 2.2. Prevalence of Plasmid-Mediated Quinolone Resistance (PMQR) Determinants among Enterobacteriaceae Strains

A total of 435 *Enterobacteriaceae* strains were subjected to PCR screening for detecting PMQR genes. At least one PMQR determinant was present in 32 (7.35%) strains belonging solely to *E. coli* species (Table 1). The *qnr*S and *qnr*B were the most common PMQR determinants, detected in 17 (53.1%) and 12 (37.5%) of 32 *E. coli* isolates, respectively (Table 1). The *aac(6*′*)-Ib-*cr gene was found in seven isolates (21.9%). Moreover, *qnr*B and *aac(6*′*)-Ib-*cr, and *qnr*B and *qnr*S were detected in combination in two isolates each, respectively. In contrast, *qnrA*, *qnrC*, *qnrD* or *qepA* were not detected in any of the tested isolates.

### 2.3. Quinolone and Fluoroquinolone Susceptibility of Tested PMQR-Positive E. coli Isolates

*E. coli* susceptibility profiles for nalidixic acid and ciprofloxacin are summarised in Table 1. All 32 PMQR-positive clinical *E. coli* isolates were resistant to nalidixic acid. Additionally, 18 of the tested isolates were also resistant to ciprofloxacin. In isolates resistant to nalidixic acid only, the ciprofloxacin MIC range was 0.125–0.25 mg/L, in contrast to *E. coli* strains resistant to both nalidixic acid and ciprofloxacin where the ciprofloxacin MIC was ≥1.5 mg/L. Among 14 ciprofloxacin-susceptible *E. coli* isolates with ciprofloxacin MIC ranges (0.125 mg/L–0.25 mg/L), *qnrB* (*n* = 5) and *qnrS* (*n* = 11) were detected, and two isolates carried both determinants. However, in 18 ciprofloxacin-resistant *E. coli* with ciprofloxacin MIC ≥ 1.5 mg/L, three different PMQR determinants were detected including *qnrB* (*n* = 6), *qnrS* (*n* = 7), and *aac(6*′*)-Ib*-cr (*n* = 7) (Table 1). Two *E. coli* isolates with ciprofloxacin MIC ≥24 mg/L carried a combination of *qnrB* and *aac(6*′*)-Ib-*cr.

### 2.4. Whole Genome Sequencing Analysis of PMQR-Positive E. coli Strains

Data of whole genome sequencing analysis of 28 PMQR-positive *E.* *coli* isolated from rectal swabs of men undergoing transrectal prostate biopsy are presented in Table 2. Notably, nearly all genomes (except one) sequenced using the Nanopore platform were assemble into one large circular chromosome with few plasmids, which were usually also circular. For these strains, the chromosome size ranged from 4,544,229 bp to 5,258,810 bp (with an average of 4,895,805 bp). All these strains had 1–13 plasmids (with an average of five plasmids) and on average one of them was large (>100 kb). Among all 28 PMQR-positive *E. coli* tested, seven (IncFI, IncFII, IncX, IncI, IncY, IncN, Col types) plasmid replicons were identified with different frequencies (Table 2).

Hypothetical pathogenic potential of analysed strains were confirmed by the PathogenFinder tool, with minimal and maximal probability of being a human pathogen estimated at 0.919 and 0.943 respectively (with an average of 0.93).

All genomes were deposited in the GenBank database under BioProject no PRJNA861130.

*Genotypic analysis of fluoroquinolone resistance.* Point mutations involving amino acid substitutions among PMQR-positive *E. coli* were detected in 21 isolates out of 28 sequenced (Table 2). The substitutions were observed in two codons of GyrA: 83 (Ser → Leu) and 87 (Asp → Asn); in three codons of ParC: 56 (Ala → Thr), 80 (Ser → Ile), and 84 (Glu→Gly, Val); in three codons of ParE: 416 (Leu → Phe), 458 (Ser → Ala) and 529 (Ile→Leu), respectively. No substitution was observed in the GyrB subunit in any of the isolates tested. The Ser83 → Leu mutation in the GyrA was the most common in PMQR-positive *E. coli* isolates, and was found in 20 isolates with a ciprofloxacin MIC at 0.125 → 32 mg/L, including five isolates of 13 susceptible to ciprofloxacin with ciprofloxacin MIC < 0.5 mg/L. The second most widespread *gyrA* mutation was Asp87 → Asn, found among 11 isolates. In *parC*, the most frequent mutation was Ser80→Ile, found in 12 isolates. Other identified mutations were observed infrequently. The PMQR-positive *E. coli* isolates resistant to ciprofloxacin with MIC > 4 mg/L had two or more substitutions, found in GyrA, ParC, and ParE (Table 2). Among genes encoding plasmid-mediated FQ resistance, *qnrS*1, *qnrB*19, and *aac(6′)-Ib-*cr were identified in 12, two, and three tested isolates, respectively.

*Analysis of genes predicted to confer resistance to other antibiotics.* According to the WGS analysis shown, 37 different determinants of resistance were identified (Table 2). Among genes encoding β-lactam resistance were detected *bla*_TEM-1B_ in 14 isolates, *bla*_OXA-1_ in three isolates, *bla*_TEM-CTX-M-15_ in two isolates, and *bla*_TEM-1C_, *bla*_TEM-1D_, *bla*_TEM-84_, *bla*_TEM-30_, *bla*_TEM-32_, *bla*_TEM-135_ in one isolate each. Seven genes encoding resistance to aminoglycosides were detected, including *aph(3*′′*)-Ib* (*n* = 7 isolates), *aph(6)-Id* (*n* = 7 isolates), *aad*A1 (*n* = 5 isolates), *aad*A5 (*n*= 4 isolates), *ant(3′′)-Ia* (*n* = 2 isolates), *aph(3′)-Ia* (*n* = 1 isolate) and *ant(3*′′*)-Ia* (*n* = 1 isolate). Genes encoding sulphonamides resistance totalled nine for *sul*1, eight for *sul*2 and one for *sul*3 and trimethoprim, five for *drf*A1, two for *drf*A17, and one each for *drf*A7 and *drf*A15. Three genes encoding resistance to macrolides, namely *mdf*A, *mph*A, and *mph*B were identified in 11, three and one isolate, respectively. Tetracycline resistance encoded by *tet*A and *tet*B genes was identified in 11 and 5 isolates, respectively. Genes encoding chloramphenicol resistance numbered three for *cat*B3, two for *cml*A1, and one each for *cat*A1 and *flo*R. Moreover, in 17 of the *E. coli* isolates tested, an operon *sit*ABCD encoding a member of the family of ATP-binding cassette divalent metal ion transporters was identified.

*Virulence genes analysis.* At least one virulence gene was detected among all sequenced *E. coli* isolates (Table 2). Table 3 summarises the distribution of 53 different virulence genes identified in 28 PMQR-positive *E. coli* samples, both sensitive and resistant to ciprofloxacin. No significant differences were found in the occurrence of virulence factors among the resistant and susceptible strains of *E. coli* tested. All tested *E. coli* isolates possessed *gad* (glutamate decarboxylase). The other most frequent virulence genes, identified among ≥10 *E. coli* sequenced isolates, were *terC* (tellurium ion resistance protein), *iss* (increased serum survival), *sitA* (iron transport protein), *ompT* (outer membrane protease-protein protease 7), *traT* (outer membrane protein complement resistance), *fyuA* (siderophore receptor), *irp2* (high molecular weight protein 2 non-ribosomal peptide synthetase), *iucC* (aerobactin synthetase), *iutA* (ferric aerobactin receptor), *iroN* (enterobactin siderophore receptor protein), and *hlyF* (hemolysin F) (Table 3).

*MLST and serotyping.* Out of the total of 28 *E. coli* isolates, 19 different STs were identified (Table 2). *E. coli* most often belonged to ST131, observed in four out of 28 tested isolates, and each of ST88, ST162, ST428, ST10, and ST46 was observed in two *E. coli* isolates. Four *E. coli* belonged to ST131, of serotypes O25:H4 (*n* = 3) and O16:H5 (*n* = 1), respectively. *E. coli* in ST88, ST428, ST10, and ST46 had the serotypes O8:H17, O117:H4, O101:H9, and O9:H10, respectively. ST162 in two *E. coli* had a different serotype i.e., O9:H19 and O33:H25. Each of ST69, ST38, ST82331, ST2207, ST1125, ST683, ST2064, ST206, ST224, ST117, ST90, ST95, and ST617 was observed in one of the 28 *E. coli* tested, respectively.

## 3. Discussion

The genetic profiling of clinical strains can provide useful information about their potential for causing disease and resistance to treatment. This study investigated the prevalence of PMQR determinants and the WGS of PMQR-positive *Enterobacterales* strains isolated from men undergoing prostate biopsy procedures, collected at 12 urological departments in Poland. These isolates exhibited a diverse range of serotypes, sequence types (ST), virulotypes, and antimicrobial-resistant gene profiles.

In general, current data shows the prevalence of PMQR determinants in 7.35% of *Enterobacterales* isolates, with some *qnrB* and *qnrS* genes found in *E. coli* isolates susceptible to FQs (ciprofloxacin MICs 0.125–0.25 mg/L). These findings correspond with the popular hypothesis that PMQR determinants may promote mutations in the quinolone resistance-determining region (QRDR) and, consequently, increase quinolone resistance in clinical settings, so that their presence may significantly contribute to the occurrence of post-biopsy complications [8,9,10]. Moreover, the findings strongly confirm the need to perform microbiological diagnostic tests (including PCR on PMQRs) before a biopsy, which has also been suggested by other authors [13,14,15].

According to the European Association of Urology (EAU), it is highly recommended to use antimicrobial prophylaxis in men prior to a transrectal prostate biopsy (TRUS-Bx), to minimise the risk of bacterial infection after the procedure [5]. All patients in our study from whom PMQR-producing isolates were collected received ciprofloxacin orally 2 h prior to the procedure. It should be noted that the PMQR resistance traits found in FQ-susceptible *E. coli* strains from patients receiving ciprofloxacin prophylaxis prior to TRUS-Bx indicated that FQ prophylaxis or treatment of post-biopsy complications may be ineffective in patients who carry PMQR-positive strains, and in consequence may promote development of high-level FQ resistance.

In our data, *qnr*S1 was dominant (78.6%) among ciprofloxacin-susceptible *E. coli* isolates, while *qnrB* was detected in 35.7% of such isolates. In the ciprofloxacin-resistant *E. coli* group with ciprofloxacin MIC ≥ 1.5 mg/L, *qnrB* and *aac(6*′*)-Ib-*cr determinants were dominant and were the same types of PMQRs reported for FQ resistant strains in other reports, [10,16]. These differences in proportions of different *qnrs* may provide additional premises for two probable means of selection of PMQR-positive strains playing essential roles in acquisition of quinolone resistance—foodborne and environmental (*qnrS*), or medical care (*qnrB*, *aac(6*′*)-Ib-*cr). Studies have suggested that the agricultural use of antimicrobial agents increased the number of human infections caused by drug-resistant bacteria [17]. Furthermore, consumption of food contaminated by PMQR-positive strains or use of FQs in ambulatory and hospital therapy may promote pre-selection factors for FQ resistance [12].

In many studies of the causes of post-biopsy complications, collections of FQ-resistant isolates have been investigated [1,2,3,18]. In those studies, FQ-resistant *E. coli* were found to occur frequently in strains isolated from men undergoing TRUS-Bx. In this prospective study, the only selection criterion was prevalence of PMQR. However, we found that 82 (18.8%) of the investigated *E. coli* strains were resistant to ciprofloxacin, with MIC > 0.5 mg/L. Differences in ciprofloxacin-resistance rates between patients from 12 urological centres were observed. The ciprofloxacin MICs for the 82 selected ciprofloxacin-resistant *E. coli* were in the range 0.75–>32 mg/L. The vast majority (*n* = 44; 53.7%) of ciprofloxacin-resistant isolates had ciprofloxacin MIC > 32 mg/L (data not shown).

WGS showed a huge variation among *E. coli* strains tested, as assessed by multi-locus sequence typing (MLST) and serotyping. In the present study, *E. coli* strains O25:H4 ST131 were found to predominate among other 19 STs detected, as shown in Table 2. One strain ST131 had a different serotype within it i.e., O16:H5. However, ST131 *E. coli* isolates of serotype O16:H5 have also been identified in Japan, Denmark, Australia, and France [19,20,21,22]. Nowadays, ST 131 has been identified as the predominant *E. coli* lineage among extraintestinal pathogenic *E. coli* (ExPEC) isolates worldwide. *E. coli* ST131 isolates have been reported to produce extended-spectrum β-lactamases (ESBL), such as CTX-M-15 or non-ESBL-produce; almost all are resistant to fluoroquinolones and ampicillin/amoxicillin, and are often resistant to aminoglycosides, macrolides, chloramphenicol and tetracyclines [23]. In our study, three isolates of *E. coli* ST131 serotype O25:H4 showed high-level resistance to ciprofloxacin (MIC > 32 mg/L) and had at least four amino acid substitutions in GyrA and ParC. Additionally, those isolates contained the *aac(6*′*)-Ib-*cr PMQR gene, CTX-M-15 (ESBL), OXA-1 conferring to resistance to ampicillin/amoxicillin, and *mph*A conferring resistance to macrolide. Isolate no. F078 *E. coli* ST131 O16:H5 was susceptible to ciprofloxacin (MIC 0.25 mg/L), but had substitutions in GyrA and ParE and possessed genes conferring to resistance to aminoglycosides (*aad*A, *aph(6)-Id*, *aph(3*′′*)-Ib*), trimethoprim (*drf*A), sulphonamides (*sul*1, *sul*2), tetracyclines (*tet*A), macrolides (*mph*A, *mdf*A), and *qnrS* detected by PCR. It should be highlighted that the multidrug-resistant ST131 clonal group was found to account for 70% of FQ-resistant rectal *E. coli* isolates among men undergoing transrectal prostate biopsy in the USA [24].

Other STs frequently detected in our study were ST88 (serotype O8:H17), ST162, ST428 (serotype O117:H4), ST10 (serotype O101:H9) and ST46 (serotype O9:H10). Each was found in two isolates, although ST162 had different serotypes i.e., O9:H19 and O33:H25. It should be noted that, in addition toST131, ST69, ST38, ST88, ST117, ST10, ST95, and ST617 are also major STs globally noted among ExPEC isolates [25].

In general, the potential of antimicrobial resistance in *E. coli* isolates tested was found to be significant. Based on the identified resistance genes, the majority of tested isolates contained at least one determinant conferring resistance. The most frequently identified genes were *bla*_TEM_, conferring resistance to β-lactams. Resistance to ampicillin/amoxicillin in *E. coli* isolates is widespread and may be mediated by production of plasmid-encoded TEM-1/2, SHV-1, or OXA-1 enzymes [23]. In our study, 16 of 28 *E. coli* isolates possessed the *bla*_TEM-1_ gene. Additionally, in four isolates we identified other *bla*_TEM_ genes encoding β-lactamase i.e., *bla*_TEM-84_, *bla*_TEM-30_, *bla*_TEM-32_, *bla*_TEM-135_. Moreover, as shown in Table 2, genes conferring resistance to aminoglycosides (*aph(3*′′*)-Ib*, a*ph(6)-Id*, *aad*), tetracyclines (*tet*A, *tet*B), macrolides (*mdf*, *mph*), sulphonamides (*sul*), and trimethoprim (*drf*) were common in the tested strains.

*E. coli* is a heterogeneous species that encompasses strains with various degrees of pathogenicity as well as non-pathogenic strains. The performed analysis confirmed that all the analyzed strains had theoretical pathogenic potential. Pathogenic *E. coli* are divided into two major groups, associated with intestinal and extra-intestinal diseases [26,27]. The extra-intestinal *E. coli* pathotypes cause urinary tract infections, neonatal meningitidis and septicemia [26]. The current study also characterised virulence genes in all PMQR-positive *E. coli* isolates from men undergoing prostate biopsy. Among 53 different virulence genes identified in all tested *E. coli* isolates, the highest frequencies were *gad*, *terC*, *iss*, *sitA*, *ompT*, *traT*, *fyuA*, *irp2*, *iucC*, *iutA*, *iroN*, and *hlyF*, some of which have been reported in pathogenic *E. coli* isolates [28]. However, the presence of single or multiple virulence genes does not confirm that a strain is pathogenic, unless that strain has the appropriate combination of virulence genes to cause infection in a host [29]. To cause infection, pathogenic *E. coli* isolates use a complex multistep mechanism of pathogenesis involving a number of virulence factors which consist of attachment, host cell surface modification, invasion, a variety of toxins, and secretion systems [30]. Thus, virulence factors are ideal target for determining the pathogenic potential of a given *E. coli* isolate [31]. Based on this finding, routine screening of bacteria isolated from men undergoing prostate biopsy can improve clinical significance.

It is worth noting that in this study, among 17 *E. coli* isolates tested, an operon sitABCD was identified encoding a member of the family of ATP-binding cassette divalent metal ion transporters. The operon sitABCD is homologous to avian pathogenic *E. coli* isolate, and mediates transport of iron and manganese as well as resistance to hydrogen peroxide. *Starowska* et al. [32] indicated the possibility of replacement of virulence genes located on mobile genetic elements between human and avian pathogenic *E. coli* strains, and that with them could be transferred genes encoding resistance to antimicrobials, ions, and other chemical components.

A limitation of our research was the performance of WGS analysis four years after susceptibility testing and PCR. This could have significantly influenced on the observed lack of correlation between the PMQR genes detected by the PCR method and the WGS analysis. *qnr* genes are localized on mobile elements and their presence can be affected by over-stabilizing the DNA-helicase complexes. Thus, they could be lost during laboratory manipulations without antimicrobial pressure. Such a situation has also been noticed by other scientists (personal communication during ECDC/EFSA fourth joint meeting on AMR for the FWD-Network and EURL-AR Network). This remains a hypothesis, and its verification needs further investigation.

In conclusion, by using whole genome sequencing, we obtained valuable information about bacterial strains and demonstrated that WGS can be a valuable early warning system for emerging resistance mechanisms and potential therapeutic failure. Our data also suggest that antimicrobial resistance is a potentially more important trait than virulence in terms of increased risk of post-biopsy infection. This is in line with the findings of other authors [33]. Meanwhile, the molecular characteristics including virulence and presence of antimicrobial resistance genes show the considerable potential of the investigated *E. coli* in the development of infection.

## 4. Materials and Methods

### 4.1. Bacterial Strains, Antimicrobial Susceptibility Testing and PCR

A prospective study was conducted between April and December 2016 at 12 urological departments in Poland. Ethics approval was obtained from the local ethics committee of the National Institute of Public Health National Institute of Hygiene (Research Ethics Committee No. 4/2015 from 2 December 2015), Warsaw, Poland. Rectal swabs were collected from 621 patients undergoing transrectal prostate biopsy. The samples were suspended in BHI broth with 20% glycerol, frozen at −20 °C and transported frozen to our laboratory. Rectal swabs were streaked onto a MacConkey agar plate (Oxoid Ltd., Basingstoke, UK) and were incubated at 37 °C for 24 h in aerobic conditions. From each plate with gram-negative bacterial growth, a representative colony of each distinct morphotype was selected for species identification by classical biochemical tests. Screening for *qnrA*, *qnrB*, *qnrS*, *aac(6*′*)-Ib-cr*, and *qepA* genes as more common PMQRs was carried out for all selected strains by using PCR as described previously [10].

All PMQR-positive isolates were screened with ciprofloxacin (5 μg) and nalidixic acid (30 μg) (Oxoid Ltd., Basingstoke, UK) on Mueller-Hinton agar (Oxoid Ltd., Basingstoke, UK). In addition, the E-test (bioMerieux, Marcy l‘Etoile, France) method was used in all PMQR-positive isolates to determine the minimum inhibitory concentration (MIC) of ciprofloxacin. The susceptibility results were interpreted according to European Committee on Antimicrobial Susceptibility (EUCAST) criteria (http://eucast.org, accessed on 1 January 2022), and a zone of inhibition around the ciprofloxacin disc (<22 mm) along with MIC of ciprofloxacin >0.5 mg/L were classified as resistance to FQs. The MIC for each isolate was measured at least twice.

### 4.2. Whole Genomes Sequencing (WGS)

Twenty-eight of the PMQR-positive isolates recovered in this study were subjected to whole genome sequencing analysis (WGS). All steps of the WGS (DNA purification, library preparation and sequencing) were performed at the National Institute of Public Health NIH—National Research Institute. Libraries for 13 strains were prepared using an Illumina DNA Prep Kit (Illumina Inc., San Diego, CA, USA). The libraries were sequenced on an Illumina MiSeq sequencer (Illumina Inc.) using 150-bp paired-end reads. Libraries for another 15 strains were prepared using a Nanopore Rapid Barcoding Kit, and the whole genome sequencing was performed on GridION (Oxford Nanopore Technologies, Oxford, UK). Genomes were assembled using CLC Genomics Workbench 22 (for strains sequenced on Illumina) and the NanoForms server (for strains sequenced on Nanopore [34]). Species confirmation, multilocus sequence typing (MLST), serotyping, virulence and AMR (Antimicrobial Resistance) genes, and plasmid replicons were analysed using SpeciesFinder 2.0 [35], MLST 2.0 [36], SerotyperFinder 2.0 [37], VirulenceFinder 2.0 [38], ResFinder 4.1 [39], respectively, available on the Centre for Genomic Epidemiology website (http://www.genomicepidemiology.org/, accessed on 1 January 2022). Additionally, the hypothetical pathogenic potential was estimated by proteome analysis using PathogenFinder 1.1 [40].

## Figures and Tables

**Table 1 ijms-23-08907-t001:** Distribution of plasmid-mediated quinolone resistance (PMQRs) determinants among different FQ-resistant phenotypes of 32 *E. coli* strains isolated from rectal swabs of men undergoing a transrectal prostate biopsy.

No.	Isolate No.	Species	Resistant to Quinolones		MIC of CIP (mg/L)	PMQRs Genes
			NA	CIP		
1	G008	*E. coli*	R	S	0.125	*qnrB*
2	A021	*E. coli*	R	S	0.19	*qnrS*
3	D025	*E. coli*	R	S	0.19	*qnrS*
4	J026	*E. coli*	R	S	0.19	*qnrB*
5	G001	*E. coli*	R	S	0.19	*qnrB*, *qnrS*
6	Ł029	*E. coli*	R	S	0.19	*qnrS*
7	Ł031	*E. coli*	R	S	0.19	*qnrS*
8	A079	*E. coli*	R	S	0.25	*qnrS*
9	A083	*E. coli*	R	S	0.25	*qnrB*
10	D008	*E. coli*	R	S	0.25	*qnrS*
11	F078	*E. coli*	R	S	0.25	*qnrS*
12	J006	*E. coli*	R	S	0.25	*qnrS*
13	L018 *	*E. coli*	R	S	0.25	*qnrS*
14	L047	*E. coli*	R	S	0.25	*qnrB*, *qnrS*
15	B047	*E. coli*	R	R	1,5	*qnrS*
16	G021	*E. coli*	R	R	2	*qnrS*
17	H015	*E. coli*	R	R	3	*qnrS*
18	G019	*E. coli*	R	R	4	*qnrS*
19	Ł036 *	*E. coli*	R	R	4	*qnrB*
20	H007 *	*E. coli*	R	R	12	*qnrB*
21	G022	*E. coli*	R	R	24	*qnrB*, *aac(6*′*)-Ib-cr*
22	G030	*E. coli*	R	R	32	*qnrB*
23	G032	*E. coli*	R	R	>32	*qnrB*, *aac(6*′*)-Ib-cr*
24	E005	*E. coli*	R	R	>32	*aac(6*′*)-Ib-cr*
25	H038 *	*E. coli*	R	R	>32	*aac(6*′*)-Ib-cr*
26	H060	*E. coli*	R	R	>32	*aac(6*′*)-Ib-cr*
27	G051	*E. coli*	R	R	>32	*aac(6*′*)-Ib-cr*
28	G053	*E. coli*	R	R	>32	*aac(6*′*)-Ib-cr*
29	H003	*E. coli*	R	R	>32	*qnrB*
30	Ł008	*E. coli*	R	R	>32	*qnrS*
31	Ł024	*E. coli*	R	R	>32	*qnrS*
32	L044	*E. coli*	R	R	>32	*qnrS*

NA—nalidixic acid, CIP—ciprofloxacin, MIC—minimum inhibitory concentration, *—*E. coli* strains not subjected to whole genome sequencing (WGS) analysis.

**Table 2 ijms-23-08907-t002:** The whole genome sequencing (WGS) analysis of 28 PMQR-producing *E. coli* strains isolated from rectal of men undergoing a transrectal prostate biopsy.

Strain No.	Species	Serotype	MLST		MIC of CIP (mg/L)	WGS Results						
						Plasmid Profile	Virulence Genes	Resistance Genes	Amino Acid Changes in:			
			Achtman	Pasteur					GyrA	GyrB	ParC	ParE
G008	*E. coli*	O15:H-	69	3	0.125	IncFIA, IncFIB	*afaA*, *afaB*, *afaC*, *afaD*, *afaE8*, *air*, *chuA*, *eilA*, *espP*, *fyuA*, *gad*, *iha*, *ireA*, *irp2*, *iss*, *iucC*, *iutA*, *kpsE*, *kpsMIII_K96*, *lpfA*, *mchB*, *mchC*, *mchF*, *ompT*, *sitA*, *terC*, *traT*	aph(3′)-Ia, aph(3′′)-Ib, blaTEM-1B, sul2, tet(B), sitABCD, mdf (A)	S83L	-	-	-
A021	*E. coli*	O86:H18	38	8	0.19	Col156, IncFIA, IncFIB, IncFII	*afaD*, *chuA*, *eilA*, *fyuA*, *gad*, *irp2*, *iss*, *iucC*, *iutA*, *kpsE*, *kpsMII_K5*, *senB*, *sitA*, *terC*, *traT*	ant(3′’)-Ia, aph(6)-Id, blaTEM-1B, sul2, tet(B), sitABCD	S83L	-	-	-
D025	*E. coli*	O8:H17	88	74	0.19	Col8282, IncFIB, IncFII, IncX1	*cia*, *cvaC*, *etsC*, *fyuA*, *gad*, *hlyF*, *iroN*, *irp2*, *iss*, *iucC*, *iutA*, *lpfA*, *mchF*, *ompT*, *papC*, *sitA*, *terC*, *traT*	qnrS1, blaTEM-1B, blaTEM-1C, tetA, sitABCD	-	-	-	-
J026	*E. coli*	O8:H17	82331	83	0.19	Col440II, IncFIB	*capU*, *cba*, *cma*, *cvaC*, *fyuA*, *gad*, *hlyF*, *ireA*, *iroN*, *irp2*, *iss*, *iucC*, *iutA*, *kpsMII_K5*, *mchF*, *ompT*, *sitA*, *terC*, *tsh*	sitABCD, mdfA	S83L	-	-	-
G001	*E. coli*	O117:H42	2207	466	0.19	IncFIB, IncI2, IncX1, IncY	*gad*, *iss*, *ompT*, *terC*	qnrS1, tetA	-	-	-	-
Ł029	*E. coli*	O55:H42	1125	302	0.19	IncFIB, ncFII, IncI-1, IncX1	*cia*, *fyuA*, *gad*, *irp2*, *iss*, *lpfA*, *ompT*, *terC*, *traT*	qnrS1, blaTEM-1B	-	-	-	-
Ł031	*E. coli*	O8:H25	unknown	24	0.19	-	*fyuA*, *gad*, *irp2*, *iss*, *lpfA*, *ompT*, *sitA*, *terC*	sitABCD	S83L	-	-	-
A079	*E. coli*	O100:H25	683	21	0.25	IncFIB, IncFII, IncI, IncX1	*cma; gad; iroN; iss; lpfA*	qnrS1, aadA1, blaTEM-1B, mdf(A), sul1, tat(A), dfrA1	-	-	-	-
A083	*E. coli*	O52:H45	2064	unknown	0.25	Col, Col156, Col8282, IncFIB, IncFIC	*gad*	qnrB19, mdf(A), tetB	-	-	-	-
D008	*E. coli*	O60:H20	206	999	0.25	IncN, IncX4	*astA*, *gad*, *terC*	qnrS1, aph(3′’)-Ib, aph(6)-Id, sul2	-	-	A56T	-
F078	*E. coli*	O16:H5	131	506	0.25	IncFIB, IncFII	*chuA*, *fyuA*, *gad*, *iha*, *irp2*, *kpsE*, *kpsMII_K5*, *ompT*, *papA_F43*, *senB*, *terC*, *traT*, *yfcV*	aadA5, aph(3′′)-Ib, aph(6)-Id, dfrA17, sul1, sul2, tet(A), sitABCD, mph(A), mdf (A),	S83L	-	-	I529L
J006	*E. coli*	O8:H17	88	74	0.25	IncX1	*fyuA*, *gad*, *irp2*, *iss*, *lpfA*, *ompT*, *papC*, *terC*	qnrS1, blaTEM-1B	-	-	-	-
L047	*E. coli*	O91:H23	224	479	0.25	IncFIB, IncFII, IncX1, IncFIC(FII), IncI1-I(Alpha)	*cia*, *cib*, *cvaC*, *etsC*, *gad*, *hlyF*, *iroN*, *iss*, *iucC*, *iutA*, *lpfA*, *mchF*, *ompT*, *sitA*, *terC*, *traT*, *tsh*	qnrB19, aadA1, aph(3′′)-Ib, aph(6) Id, blaTEM-1B, dfrA1, sitABCD, sul1, sul2, tet(A), floR	-	-	-	-
B047	*E. coli*	O9:H19	162	294	1.5	Col8282, IncFII, Inc1-I, IncX4	*capU*, *gad*, *iss*, *ipfA*, *ompT*, *terC*, *traT*	qnrS1, blaTEM-1B	S83L	-	-	-
G021	*E. coli*	O117:H4	428	73	2	IncFIB, IncX1, IncFIC (FII)	*chuA*, *cma*, *etsC*, *fyuA*, *gad hlyF*, *hra*, *ibeA*, *iroN*, *irp2*, *iss*, *iucC*, *iutA*, *kpsE*, *kpsMII_K5*, *neuC*, *ompT*, *sitA*, *terC*, *traT*, *tsh*, *usp*, *vat*, *yfcV*	qnrS1, blaTEM-1B, sitABCD	S83L	-	-	-
H015	*E. coli*	O117:H4	428	73	3	IncFIB, IncX1	*cma; gad; iroN; iss*	qnrS1, blaTEM-1B, mdf(A)	S83L	-	-	-
G019	*E. coli*	O33:H4	117	48	4	IncFIB, IncFIB	*astA*, *cea*, *chuA*, *etsC*, *fyuA*, *gad*, *hlyF*, *hra*, *ireA*, *iroN*, *irp2*, *iss*, *iucC*, *iutA*, *lpfA*, *ompT*, *pic*, *sitA*, *terC*, *traT*, *vat*	ant(3′′)-Ia, blaTEM-84, dfrA1, sul1, catA1, sitABCD	S83L D87N	-	S80I	-
G022	*E. coli*	O33:H25	162	355	24	not found	*gad; lpfA*	mdf(A)	S83L D87N	-	S80I E84G	-
G030	*E. coli*	O8:H9	90	66	32	Col440II, IncFIB, IncFII	*cma*, *cvaC*, *gad*, *hlyF*, *iroN*, *iss*, *ompT*, *sitA*, *terC*, *traT*	aadA1, aph(3′′)-Ib, aph(6)-Id, blaTEM-1B, dfrA1, sul1, sul2, tet(A), sitABCD, mdfA	S83L D87N	-	S80I	S458A
G032	*E. coli*	O101:H9	10	unknown	>32	IncFIB, IncFII	*cma*, *cvaC*, *gad*, *hlyF*, *iroN*, *iss*, *ompT*, *sitA*, *terC*, *traT*	aadA1, aph(3′′)-Ib, aph(6)-Id, blaTEM-1B, dfrA1, sul1, sul2, tet(A), sitABCD, mphB	S83L D87N	-	S80I	L416F
E005	*E. coli*	O25:H4	131	43	>32	IncFIA, IncFII	*gad; iha; iss; nfaE; sat*	aac(6′)-Ib-cr, aadA5, blaCTX-M-15, blaOXA-1, mphA, catB3, sul1, tet(A), dfrA17	S83L D87N	-	S80I E84V	I529L
H060	*E. coli*	O101:H9	10	2	>32	IncFIA, IncFIB, IncX1	*gad*, *iucC*, *iutA*, *papC*, *sitA*, *terC*, *traT*	blaTEM-30, dfrA7, sul1, sul2, tet(B), sitABCD, mdfA	S83L D87N	-	S80I	L416F
G051	*E. coli*	O25:H4	131	43	>32	Col156, IncFIA, IncFIB, IncFII	*chuA*, *cnf1*, *fyuA*, *gad*, *hra*, *iha*, *irp2*, *iss*, *iucC*, *iutA*, *kpsE*, *kpsMII_K5*, *ompT*, *papA_F43*, *papC*, *sat*, *senB*, *sitA*, *terC*, *traT*, *usp*, *yfcV*	aac(6′)-Ib-cr, blaOXA-1, sitABCD, catB3	S83L D87N	-	S80I E84V	I529L
G053	*E. coli*	O25:H4	131	43	>32	Col156, IncFIA, IncFIB, IncFII	*celB*, *chuA*, *fyuA*, *gad*, *hra*, *iha*, *irp2*, *iss*, *iucC*, *iutA*, *kpsE*, *kpsMII_K5*, *ompT*, *papA_F43*, *papC*, *sat*, *senB*, *sitA*, *terC*, *traT*, *usp*, *yfcV*	aac(6′)-Ib-cr, aadA5, blaCTX-M-15, blaOXA-1, sitABCD, mphA, sul1, catB3	S83L D87N	-	S80I E84V	-
H003	*E. coli*	O1:H7	95	1	>32	IncFIB, IncFII	*chuA*, *cia*, *cvaC*, *etsC*, *fyuA*, *gad*, *hlyF*, *ireA*, *iroN*, *irp2*, *iss*, *iucC*, *iutA*, *kpsE*, *kpsMII_K1*, *mchF*, *neuC*, *ompT*, *papA_F11*, *papC*, *sitA*, *terC*, *traT*, *usp*, *vat*, *yfcV*	blaTEM-1D, sitABCD, tet(A)	S83L	-	S80I	-
Ł008	*E. coli*	O9:H10	46	736	>32	IncFIB, IncFII, IncX1, IncFIC (FII), IncI1-I(Alpha)	*cib*, *cma*, *cvaC*, *fyuA*, *gad*, *hlyF*, *iroN*, *irp2*, *iss*, *ompT*, *sitA*, *terC*, *traT*	qnrS1, aadA1, blaTEM-1B, dfrA15, sitABCD, sul3, tet(A), cmlA1	S83L D87N	-	S80I	S458A
L024	*E. coli*	O9:H10	46	736	>32	IncX1	*fyuA*, *gad*, *irp2*, *terC*	qnrS1, blaTEM-1B, mdfA	S83L D87N	-	S80I	S458A
L044	*E. coli*	O101:H10	617	2	>32	IncFIA, IncFIB, IncFII, IncX4, IncN	*capU*, *gad*, *iss*, *sitA*, *terC*, *traT*	qnrS1, aadA5, ant(2′′)-Ia, aph(3′′)-Ib, aph(6)-Id, blaTEM-135, blaTEM32, sitABCD, tet(A), tet(B), mdfA	S83L D87N	-	S80I	S458A

**Table 3 ijms-23-08907-t003:** Distribution of virulence genes in 28 PMQR-producing *E. coli* strains isolated from rectal swabs of men undergoing a transrectal prostate biopsy.

Virulence Factor	Virulence Function	Virulence Gene	Total	CIP S MIC ≤ 0.25 mg/L	CIP R MIC >0.5 mg/L
Gluta mate decarboxylase	enables the bacteria to survive under acidic conditions	*gad*	28	14	14
Tellurium ion resistance protein	involved in tellurite resistance by catalysing efflux of tellurium ions; in complex with TerB protein manages membrane stress response	*terC*	22	12	10
Increased serum survival	involved in resistance process against serum complement system; the protection factor against phagocytosis	*iss*	17	6	11
Iron transport protein	iron ion transport	*sitA*		7	10
Outer membrane protease (protein protease 7)	confers *E. coli* resistance to the antimicrobial activity of urinary cationic peptides, including defensins	*ompT*	16	8	8
Outer membrane protein complement resistance	reduces the sensitivity of *E. coli* cells to phagocytosis by macrophages; inhibition of the classical pathway of complement activity	*traT*	15	5	10
Siderophore receptor	ferric yersiniabactin uptake receptor FyuA involved in iron-acquisition mechanism; involved in biofilm formation process in iron-poor environments such as human urine	*fyuA*	14	8	6
High molecular weight protein 2 non-ribosomal peptide synthetase	maintains cellular iron homeostasis by regulating the expression of genes involved in iron metabolism	*irp2*	14	8	6
Aerobactin synthetase	involved in siderophore aerobactin synthesis	*iucC*	12	6	6
Ferric aerobactin receptor	aerobactin iron transport system	*iutA*	12	6	6
Enterobactin siderophore receptor protein	urovirulence factor mediates utilisation of the siderophore enterobactin	*iroN*	11	4	7
Hemolysin F	involved in the increased production of outer membrane vesicles (OMVs), which leads to increased release of toxins such as the cytolethal distending toxin (CDT), cytolysin A (ClyA) and autophagy of infected eukaryotic epithelial cells	*hlyF*	10	4	6
Long polar fimbriae	adhesive factors of Shiga toxin-producing *E. coli* (STEC), associated with colonisation of the intestine	*lpfA*	9	6	3
Polysialic acid transport protein; Group 2 capsule	involved in transport of polysialic acid capsule across the cytoplasmic membrane to the bacteria cell surface protects from phagocytosis	*kpsMII*	9	5	4
Outer membrane hemin receptor	responsible for heme uptake	*chuA*	8	3	5
Colicin M	cytotoxic protein responsible for murein degradation and cell lysis, specifically inhibits murein and O-antigen biosynthesis by interfering with the regeneration of bactoprenol	*cma*	8	3	5
Capsule polysaccharide export inner-membrane protein	involved in exporting capsular polysaccharides across the periplasm and outer membrane, which precense facilitate pathogen bacteria evasion of host immune responses	*kpsE*	7	3	4
Microcin C	kills competitor cells by disrupting the membrane potential	*cvaC*	7	3	4
Outer membrane usher *P. fimbriae*	responsible for the assembly and secretion of virulence factor, pilus P	*papC*	6	2	4
ABC transporter protein MchF	involved in Mcc, MccH47 siderophore transport outside the producing bacteria	*mchF*	6	5	1
Putative type I secretion outer membrane protein	efflux transmembrane transporter activity	*etsC*	5	2	3
Adherence protein	the most frequently detected adhesins among isolates from UTI patients probable role in adherence to the epithelial cells in *eae*-negative STEC strains	*iha*	5	2	3
Siderophore receptor	urovirulence factor, involved in iron acquisition	*ireA*	5	3	2
Heat-resistant agglutinin	confers adherence to eukaryotic cell surfaces and causes bacterial auto-aggregation	*hra*	5	1	4
Serine protease autotransporters of *Enterobacteriaceae*	capable of adhering to red blood cells, hemoglobin, extracellular matrix proteins fibronectin and collagen IV hemagglutination and proteolytic activity	*tsh*	4	3	1
Major pilin	enables bacteria to colonise the epithelium of specific host organs	*papA*	4	1	3
Plasmid-encoded enterotoxin	involved in the development of severe diarrhea in patients infected with Shigella and enteroinvasive *E. coli* (EIEC)	*senB*	4	2	2
Uropathogenic specific protein	bacteriocin-like genotoxin possesses DNase activity, exhibits genotoxic activity in mammalian cells when coapplied with Imu2 protein provokes DNA damage which leads to apoptosis and ageing of cells	*usp*	4	0	4
Fimbrial protein	major subunit of a putative chaperone-usher fimbria	*yfcV*	4	1	3
Hexosyltransferase homolog	probably responsible for EAEC diarrhea in younger children < 12 months	*capU*	4	2	2
Colicin ia	induced by the SOS system during stress destruction of the cell by forming pores in the affected cell’s cytoplasmic membrane	*cia*	4	3	1
Serine protease autotransporters of *Enterobacteriaceae*	cleavage of spectrin, induces cytopathic effects on epithelial cells	*sat*	3	0	3
Colicin ib	kills other competitor bacterial strains by pore formation in the inner membrane	*cib*	2	1	1
Polysialic acid capsule biosynthesis protein	an essential enzyme in the biosynthesis of sialic acid, which is a component of *E. coli* capsule	*neuC*	2	0	2
Serine protease autotransporters of *Enterobacteriaceae*	vacuolating autotransporter toxin present in some uropathogenic *E. coli* (UPEC) strains induces cellular damage, vacuole formation, urothelial barrier dysregulation of bladder epithelial cells	*vat*	2	0	2
Afimbrial adhesion	mediates the internalisation of adherent bacteria into cells	*afaD*	2	2	0
Salmonella HilA homolog	probable role in modulation expression of the TTSS and effectors of different chromosomal islands in EAEC	*eilA*	2	2	0
Colicin B	destroys sensitive bacteria cells by dispersion the membrane potential through channel formation	*cba*	2	2	0
Heat-stable enterotoxin 1	activate guanylate cyclase in the intestinal epithelial cells leading to watery diarrhea	*astA*	2	1	1
Diffuse adherence fibrillar adhesin gene	involved in diffuse adherence of *E.coli* strains to the mucosa of the small intestine	*nfaE*	1	0	1
Endonuclease colicin E2	nuclease activity, involved in cell lysis by degradation of DNA active under induction of the SOS response by e.g., DNA damage	*celB*	1	0	1
Cytotoxic necrotizing factor	by activating Rho proteins, CNF1 factor provokes actin cytoskeleton rearrangement in the human epithelial cells	*cnf1*	1	0	1
Invasin of brain endothelial cells	contributes to *E. coli* K1 invasion of the blood–brain barrier	*ibeA*	1	0	1
Transcriptional regulator	regulates the transcription of genes involved in the biosynthesis of afimbrial adhesin, Afa	*afA*	1	1	0
Periplasmic chaperone	involved in the biogenesis of afimbrial adhesin, Afa	*afaB*	1	1	0
Outer membrane usher protein	pivotal for virulence factor, Afa adhesins biogenesis	*afaC*	1	1	0
Adhesin protein	mediates the mannose-resistant hemagglutination (MRHA) of human erythrocytes and specific attachment to epithelial cells by recognition of the decay-accelerating factor (DAF) receptor	*afaE*	1	1	0
Enteroaggregative immunoglobulin repeat protein	may promote bacterial aggregation and colonisation in infected host	*air*	1	1	0
Putative exoprotein precursor	protease activity, capable to cleaving pepsin A and human coagulationfactor V, which degradation could contribute to the mucosal hemorrhage	*espP*	1	1	0
Microcin H47 part of colicin H	precursor peptide of the microcin H47, which impairs the cellular proton channel in the target cells	*mchB*	1	1	0
MchC protein	involved in microcin H47 biosynthesis	*mchC*	1	1	0
Colicin E1	leading to death of other *E.coli* cells by forming pores on the inner membrane	*cea*	1	0	1
Serine protease autotransporters of *Enterobacteriaceae*	autotransporter secreted by EAEC and other *E. coli* pathotypes; mucus secretagogue capability and mucinolytic activity in human goblet cells which secrete MUC2 and MUC5AC	*pic*	1	0	1

## Data Availability

All data are presented within the article. The authors confirmed that personal identity information of the patients was unidentifiable from this report.

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
