# Peer review of "Prevalence of Plasmid-Mediated Quinolone Resistance (PMQRs) Determinants and Whole Genome Sequence Screening of PMQR-Producing E. coli Isolated from Men Undergoing a Transrectal Prostate Biopsy"

_ijms, 2022, doi:10.3390/ijms23168907_

Round 1

Reviewer 1 Report

This study has tested the significance of relationships of FQ-resistant rectal flora in post-TRUST-Bx infections using whole genome sequencing. The finding of this study put forward a variety of serotypes and sequence types in E. coli. All strains identified were proved to be a carrier of at least one virulence gene, and different classes of antibiotic resistance genes were identified. This study’s data support the hypothesis that PMRQ-producing E. coli isolated from patients undergoing transrectal prostate biopsy are FQ resistant. Genes and mutations related to this resistance have been determined. The data obtained from this study seems sound and valuable. I believe that this study can take the attention of scientists and clinicians and would be helpful in the prevention of infections in patients who have undergone transrectal prostate biopsy. Therefore, I suggest this paper be published in IJMS after revision in accord with the comments given below:

1.       Some sentences and statements were repeated several times in different parts of the manuscript.

2.       Some of the results were also repeated in the discussion section. Please reduce such repetitions in the discussion as much as possible.

3.       A limitation of this study has been given in the discussion stating that “WGS analysis four years after the susceptibility testing and the PCR could have significantly influenced on 125 the observed lack of correlation between the PMQR genes detected in the PCR method 126 and their lack in the WGS analysis.” This statement is unclear and thus requires more clarification because it is critical for the value of the data.

1.        AMR genes are given in the abstract only and never mentioned again. What does AMR stand for? Please clarify.

2.       Enterobacterales is given sometimes in italics, sometimes not.

3.       Rectal swabs were collected from patients undergoing transrectal prostate biopsy and transported to the laboratory. Please give some details on what conditions were transferred to the laboratory.

4.       Please check the English carefully. Some sentences have some grammatical errors.

Reviewer 2 Report

Major issues:

Extremely bad English. Extensive editing and rewriting is needed to make the paper legible,

Abstract – there is no definition of PMQR in the abstract, it is instead in the title and it would have been better the other way around.

"The 28 of 32 strains producing PMQRs was subjected for WGS" – this sentence is not clear and need to be rewritten

"The variety serotypes and sequencetypes (STs) of E. coli were notice." – not noticed – determined or checked – notice is not the correct term.

"…at least one of virulence genes – virulence gene not genes

AMR genes encoded resistance against different classes of antibiotics – why is AMR not explained? This is badly written again

"… what partially may explain cause…" – that partially not what.

"genes detected shows a considerable potential of E. coli tested in develop infection." – this is so badly worded it is impossible to guess what you meant to say

" due to widespread emergency" – you mean emergence not emergency

"… The studies suggest that" – remove the 'The' it is redundant

"… since over two decades" – please replace the since here it is not appropriate.

"… Recently, our team published study on the molecular mechanisms" – published a study

What MLST typing did you use? Pasteur or Achtman?

" strains belonging sole to E. coli species" – what does that mean? Soley? Or did you mean something else?

"… determinants detected in 17 (53.1%) and 12 (37.5%) of 32 E. coli isolates, respectively" – the paragraph is awkwardly cut into two halves

Table 2: What does it mean that a plasmid profile is marked as 'no data'? Plasmidfinder didn’t find a plasmid? Why is the table so badly arranged with words spilling out from their cells?

" in the amount of 1 to 13 (average 5 plasmids) and on average 1 of them were large" – badly worded.

Discussion – is 7.35% a significant prevalence? If so, where are the references to show this?

" This finding support the hypothesis that PMQR determinants may promote the quinolone resistance-determining region (QRDR) mutations thus increasing quinolone resistance in clinical settings and their presence may significantly contribute to the occurrence of post-biopsy complications." – This might be true but you have not done enough to make this claim

" foodborne and environmental (qnrS) or medical care (qnrB, aac(6')-Ib-cr)." – your results do not have any relation to foodborne infections so why are you saying they provide evidence related to it?

" The WGS showed a huge" – just WGS, remove the 'The'

Minor issues:

"However, their increased use has led to increasing bacterial resistance, mainly among different species of Enterobacterales" – Enterobacterales should be in italics – this mistake is throughout the text!!!

Reviewer 3 Report

This manuscript describes investigations of fluoroquinolore resistance determinants in Escherichia coli isolates obtained from restal swabs.

In my opinion this study gives a very good overview about the fluoroquinoone resistance determinants, however, some parts of this manuscript are unclear:

Comments

1) What was the reason to check nalidix acid  and ciprofloxacin susceptiblity on these strains and not to perform susceptiblity of other fluoroquinolones?

2) Whole genome sequence data should be uploaded to any of the databases, and please indicate accession number of your sequences!

Round 2

Reviewer 2 Report

The paper is greatly improved, but I do suggest an additional read-through for finer tuning